# Realization of topological Mott insulator in a twisted bilayer graphene lattice model

Bin-Bin Chen [1,2], Yuan Da Liao[3,4], Ziyu Chen[1], Oskar Vafek[5,6], Jian Kang [7✉], Wei Li [1,8✉] &
Zi Yang Meng [2✉]

Magic-angle twisted bilayer graphene has recently become a thriving material platform realizing correlated electron phenomena taking place within its topological flat bands. Several numerical and analytical methods have been applied to understand the correlated phases therein, revealing some similarity with the quantum Hall physics. In this work, we provide a Mott-Hubbard perspective for the TBG system. Employing the large-scale density matrix renormalization group on the lattice model containing the projected Coulomb interactions only, we identify a first-order quantum phase transition between the insulating stripe phase and the quantum anomalous Hall state with the Chern number of ±1. Our results not only shed light on the mechanism of the quantum anomalous Hall state discovered at three-quarters filling, but also provide an example of the topological Mott insulator, i.e., the quantum anomalous Hall state in the strong coupling limit.

[1] School of Physics, Beihang University, Beijing 100191, China. [2] Department of Physics and HKU-UCAS Joint Institute of Theoretical and Computational Physics, The University of Hong Kong, Pokfulam Road, Hong Kong, China. [3] Beijing National Laboratory for Condensed Matter Physics, and Institute of Physics, Chinese Academy of Sciences, Beijing 100190, China. [4] School of Physical Sciences, University of Chinese Academy of Sciences, Beijing 100190, China. [5] Department of Physics, Florida State University, Tallahassee, FL 32306, USA. [6] National High Magnetic Field Laboratory, Tallahassee, FL 32310, USA. [7] School of Physical Science and Technology & Institute for Advanced Study, Soochow University, Suzhou 215006, China. [8] CAS Key Laboratory of Theoretical Physics, Institute of Theoretical Physics, Chinese Academy of Sciences, Beijing 100190, China. ✉email: jkang@suda.edu.cn; w.li@buaa.edu.cn; zymeng@hku.hk

Twisted bilayer graphene (TBG) burst on the scene as a tunable two carbon-atom layers thick system realizing a remarkable multitude of interaction-driven macroscopic quantum phenomena[1–20]. Although significant progress has been achieved in understanding the nontrivial topology of the narrow bands, as well as the correlated electron states in the magic-angle TBG[21–42], many important questions remain open. One of the most fascinating question is the origin and the mechanism of the quantum anomalous Hall (QAH) state with Chern number $C = \pm 1$[6,7] at three-quarters filling of the system, aligned with the hexagonal boron nitride (hBN), and the insulating state which replaces the QAH in devices without the hBN alignment.

Currently, the prevailing opinion is that the QAH can be obtained from narrow band models with large Coulomb interactions[28,43–46], but that the nontrivial topology of the narrow bands prevents a faithful construction of local "Hubbard-like" tight-binding models that locally respect all the symmetries[23]. Although there exists no a priori Wannier obstruction, as the narrow bands' total Chern number vanishes, there is yet no clear understanding of how the QAH could arise within such correlated lattice model, even in principle, in the limit where the Coulomb interactions dominate the kinetic energy.

Precisely such a state was sought by Raghu et al. in an entirely different context[47], coining the term topological Mott insulator (TMI), which we define to be a QAH in a strong coupling limit of a local lattice model with a vanishing ratio of the bandwidth to Coulomb interaction. However, the original proposal[47] was subsequently shown not to host a QAH, and therefore not TMI either[48,49]. More recent works have found the interaction-induced QAH state in a different model, but it is stabilized by the kinetic energy and necessitates sizable bandwidth[50–52]. Because it gives way to more conventional Mott insulators in the strong coupling regime[52], these models do not host a TMI.

Here we show that the TMI is realized in a simple lattice model introduced by two of the authors as a local description of the correlations within the TBG narrow bands[26,53,54]. The key new ingredients are the off-site terms appearing alongside the usual on-site terms in the projected density operator. Physically, such terms originate in the extended multi-peak nature of the maximally localized Wannier states[22,24] arising from the nontrivial topology[25,25,35,55–58] of the narrow bands, and, importantly, remain finite even when the bandwidth vanishes.

## Results

### Honeycomb moiré lattice model.
In the strong coupling limit, the aforementioned model (as illustrated in the upper panels of Fig. 1) is

$$H = U_0 \sum_{\hexagon} (Q_{\hexagon} + \alpha T_{\hexagon} - 1)^2, \tag{1}$$

where $U_0$ constitutes the overall energy scale in the problem ($\approx 40$ meV in TBG and set to unity henceforth). $Q_{\hexagon} \equiv \frac{1}{3}\sum_{l=1}^{6} c^{\dagger}_{\mathbf{R}+\delta_l} c_{\mathbf{R}+\delta_l}$ represents the cluster charge term[22,25,53,54,59,60] (c.f. Fig. 1c), and $T_{\hexagon} \equiv \sum_{l=1}^{6}[(-1)^l c^{\dagger}_{\mathbf{R}+\delta_{l+1}} c_{\mathbf{R}+\delta_l} + h.c.]$ represents the Coulomb induced hopping with alternating sign (c.f. Fig. 1d). Fermion annihilation and creation operators $c_{\mathbf{R}+\delta_l}$ and $c^{\dagger}_{\mathbf{R}+\delta_l}$ are defined at the sites of the honeycomb lattice $\mathbf{R} + \delta_l$, where $\mathbf{R} = m_1\mathbf{L}_1 + m_2\mathbf{L}_2$ with integer $m_{1,2}$ spans the triangular Bravais lattice. The hexagon centers, over which we sum in Eq. (1), are connected to the six nearest honeycomb lattice sites $l = 1, 2, \cdots 6$ through $\delta_l$ (c.f. Fig. 1e). As we focus on the three-quarters filling of the TBG, where the spin and orbital degrees of freedom are assumed to be polarized, Eq. (1) thus constitutes a simplification to the full Hamiltonian of ref. [26]. The parameter $\alpha$ controls the relative strength of charging and assisted-hopping of the projected Coulomb interaction. It originates from the overlap of two neighboring Wannier states in the continuum model and thus depends on the lattice relaxation. Due to the background charge from the remote bands, which is approximated to be uniform in Eq. (1), the projected Coulomb interaction is in the form of density-density repulsion[43,61,62], instead of being normal ordered. Although the projected interaction contains other terms such as next-nearest neighbor interaction, the more detailed calculations at the chiral limit have shown that the interaction-induced dispersion of the charged excitation at the charge neutrality point is dominated by $\alpha$, the nearest neighbor assisted hopping[63].

The original bandwidth $W \sim 8$ meV[24] is much smaller than $U_0$, suggesting the system is in the strong coupling regime. Furthermore, after the states on the remote bands are integrated out, the superexchange interaction ($\lesssim 5 \times 10^{-3} e^2/(\epsilon L_m)$) is found to be negligible compared with the projected Coulomb interaction[61]; this justifies neglecting additional fermion bilinear (kinetic) terms in Eq. (1). The kinetic term, as well as the further-range assisted hopping terms, may shift the critical value $\alpha_c$ of the phase transition but do not qualitatively change the phase diagram in Fig. 1f. In addition, we do not include the additional symmetry breaking term produced by the possible hBN alignment that favors the QAH phase[64], but focus on the topological phase transitions purely driven by interactions.

It is worth emphasizing that Eq. (1) corresponds to the leading order terms when the distance to the gates $l_g$ is about the same as the moiré lattice constant $|\mathbf{L}_1|$, and thus the electron-electron repulsion decays exponentially when the inter-electron separation is larger than $|\mathbf{L}_1|$[26]. With larger $l_g$, the longer range aspect of the Coulomb repulsion will have to be included, but because currently there is no experimental indication that there are significant changes in the nature the insulating states for different $l_g$[1,3,65], it is reasonable to neglect the longer range terms in Eq. (1). We should note that terms in Eq. (1) are purely real, and because the two QAH states with opposite Chern numbers transform into each other under complex conjugation, the QAH state is not a priori favored by this model. In what follows, we will demonstrate that, for a range of $\alpha$, Eq. (1) naturally leads to the TMI ground state via spontaneous symmetry breaking without including any other interactions or kinetic terms.

### Phase diagram.
We solve the TBG lattice model in Eq. (1) using DMRG on long cylinders of XC (zigzag, Fig. 1a) and YC (armchair, Fig. 1b) geometries, with widths $W$ up to 6 and lengths $L$ up to 24. The details of DMRG implementation and finite-size analysis are given in the Methods and Supplementary Note 1. The obtained ground state phase diagram, as a function of $\alpha$, is shown in Fig. 1f. We identify two gapped insulating phases: a stripe phase with charge density wave (CDW) for small $\alpha$, and a TMI phase for $\alpha > \alpha_c \approx 0.12$. These two ground states are separated by a first-order quantum phase transition (QPT). In Fig. 2, we show results for various quantities, including the ground state energy $e_g$, entanglement entropy $S_E$, charge structure factor $C_n$, and the imaginary part of the equal time correlation $\langle J \rangle \equiv \frac{i}{2}\langle (c_l^{\dagger} c_{l'} - c_{l'}^{\dagger} c_l)\rangle$. As shown in Fig. 2a, the $e_g$ curve exhibits a discontinuity in the slope (a kink) at $\alpha_c$, indicating the first-order QPT. In Fig. 2b, we calculate the entanglement entropy $S_E(x) \equiv -\text{Tr}[\rho_{\mathcal{A}}(x)\ln(\rho_{\mathcal{A}}(x))]$, with $\rho_{\mathcal{A}}(x)$ the reduced density matrix of the subsystem $\mathcal{A}$ consisting of the first $x$ columns (c.f. Fig. 1a, b). By setting $x = L/2$ (for even $L$), i.e., cutting at the very center of the system, we compute $S_E(L/2)$ and show it vs. $\alpha$ in Fig. 2b, where an evident "jump" takes place right at the QPT. In addition, for $\alpha < \alpha_c$ the negligibly small $S_E(L/2)$ indicates the existence of a nearly direct product state with virtually no charge fluctuations in the CDW pattern. On the other hand, the sizable $S_E(L/2)$

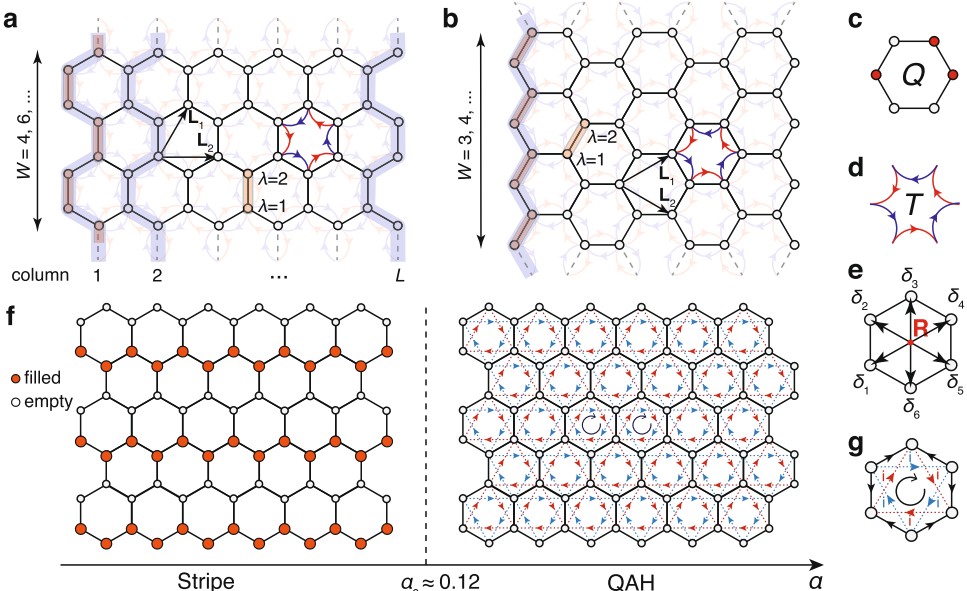

**Fig. 1 The honeycomb moiré lattice model and phase diagram. a** YC and **b** XC geometries with PBC along vertical ($\mathbf{L}_1 - \mathbf{L}_2$ for XC and $2\mathbf{L}_1 - \mathbf{L}_2$ for YC) and OBC along horizontal direction. The number of sites on the cylinders is $N = W \times L \times 2$, with length $L$ (the number of vertical armchair/zigzag chains, c.f. the gray-shaded lines) and $W$ is the number of 2-site unit cells (c.f. the red-shaded rectangles) along those chains. **c** Shows the cluster charge operator $Q_{\hexagon}$, which counts the electron number in a hexagon and **d** demonstrates the assisted hopping term $T$ with alternating-sign structure. **e** The labeling of six sites within hexagon $\mathbf{R}$. **f** The phase diagram contains two distinct insulating phases, i.e., the stripe phase for $\alpha < \alpha_c$, and the QAH state for $\alpha > \alpha_c \simeq 0.12$. **g** The schematic plot of the emergent current through a mean-field tight-binding analysis of the QAH state.

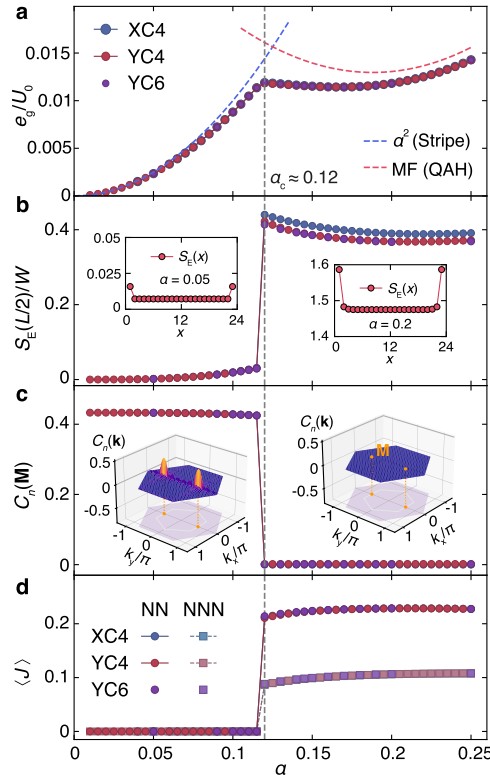

**Fig. 2 Identification of two insulating phases. a** The ground-state energy per site $e_g \equiv \frac{1}{N} \langle \psi_g | \hat{H} | \psi_g \rangle$, shown as a function of $\alpha$, with total number of sites $N = 2WL$ and $|\psi_g\rangle$ the DMRG ground state. **b** Entanglement entropy $S_E$, **c** stripe order parameter $C_n(\mathbf{M})$, **d** both correlations $\langle J \rangle_{\text{NN}}$ and $\langle J \rangle_{\text{NNN}}$, are shown versus $\alpha$, all showing abrupt changes of behavior at $\alpha_c \simeq 0.12$. The mean-field energies for both phases are as well shown in **a**. The detailed entanglement profile $S_E$ vs. subsystem $x$ is shown in the inset of (**b**), and $C_n(\mathbf{k})$ vs. **k** in the first Brillouin zone (BZ) shown in the inset of (**c**).

for $\alpha > \alpha_c$ indicates a finite amount of quantum entanglement in the ground state. In the insets of Fig. 2b, $S_E(x)$ vs. subsystem length $x$ shows a flat plateau in the bulk of the system, indicating that both phases in Fig. 1f are gapped, consistent with the exponentially decaying single-particle Green's functions also obtained by our DMRG (see the Supplementary Note 1).

**Stripe and QAH insulators**. The emergence of the stripe phase at small $\alpha$ can be understood from a perturbative analysis[26]. Up to second-order corrections (c.f. Supplementary Note 2), we find the ground-state energy $e_g/U_0 \simeq \alpha^2$, and plot it together with the DMRG results in Fig. 2a, where the high accuracy of this analytical calculation can be clearly seen. The CDW order can be characterized by the structure factor, $C_n(\mathbf{k}) \equiv \frac{1}{N} \sum_{\lambda=1}^{2} \sum_{\mathbf{R}} e^{-i\mathbf{k} \cdot (\mathbf{R}+\delta_\lambda)} \tilde{n}_{\mathbf{R},\lambda}$, where the quantity $\tilde{n}_{\mathbf{R},\lambda} = \langle c_{\mathbf{R}+\delta_\lambda}^\dagger c_{\mathbf{R}+\delta_\lambda} \rangle - 1/2$ counts the number of electrons (with respect to the half filling) on the honeycomb site $\mathbf{R} + \delta_\lambda$. In Fig. 2c, we find that $C_n(\mathbf{k})$ peaks at $\mathbf{M} = (0, \frac{2\pi}{\sqrt{3}|\mathbf{L}_1|})$ for $\alpha < \alpha_c$, and drops abruptly to 0 for $\alpha > \alpha_c$, confirming that the small-$\alpha$ regime has a CDW order, while for $\alpha > \alpha_c$ the insulating phase has no charge order. Remarkably, this $\alpha > \alpha_c$ regime turns out to be a topological phase with spontaneous time-reversal symmetry (TRS) breaking and a quantized Hall conductance, i.e., a QAH phase.

To reveal the TRS breaking in the large-$\alpha$ QAH phase, in Fig. 2d we show the correlation $\langle J \rangle$ on both the nearest-neighbor (NN) and next-nearest-neighbor (NNN) $(l, l')$ pairs. We find a finite value of $\langle J \rangle_{\text{NN}} \sim 0.22$ and $\langle J \rangle_{\text{NNN}} \sim 0.1$ in the bulk of the cylinder for large-$\alpha$ phase, while they vanish in the stripe phase. In the QAH phase, the real part of $\langle c_l^\dagger c_{l'} \rangle$ is negligibly $[O(10^{-7 \sim -8})]$ smaller compared to its imaginary part, and thus $\langle c_l^\dagger c_{l'} \rangle$ emerging from interactions is virtually purely imaginary. The corresponding hopping process thus acquires a $\pi/2$ phase (labeled as $i$ in Fig. 1g), rendering a $3\pi/2$ flux for a circulating triangular loop current, which resembles the Haldane model[66]. The difference is that the TRS breaking NNN hopping term is introduced explicitly in the Haldane model, while here it emerges spontaneously due to electron interactions, a typical feature of TMIs.

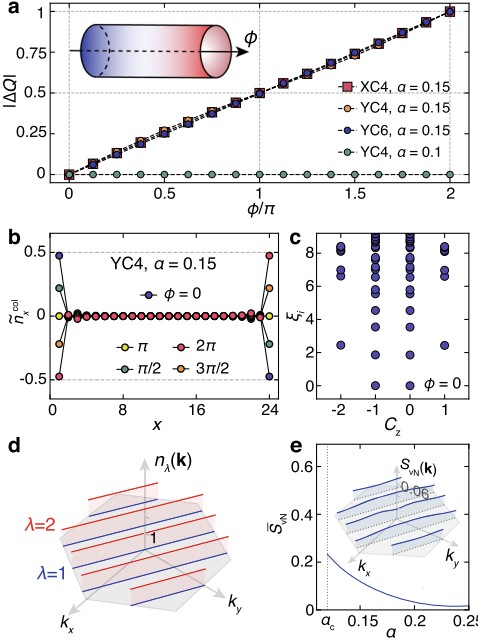

**Fig. 3 Quantized hall conductance and QAH state.** In the systems of both width $W = 4, 6$, a flux $\phi \in [0, 2\pi]$ is threading through the cylinder, **a** One electron is pumped from one edge to the other for $\alpha = 0.15$ (QAH phase), while no charge response is observed for $\alpha = 0.1$ (stripe phase). **b** In the real-space charge distribution of YC4 cylinder, no accumulations are observed in the bulk, i.e., only the charge near the left edge is pumped. **c** Entanglement spectrum computed at the central bond (between two columns) shows a two-fold degeneracy. For a typical QAH state with $\alpha = 0.25$, we show in **d** the charge density $n_\lambda(\mathbf{k})$, with $\lambda$ labeling the two eigenvalues of the $2 \times 2 \tilde{G}(\mathbf{k})$ matrix associated with two sublattices. In **e** the von Neumann Entropy $\bar{S}_{vN}$ averaged over all the $\mathbf{k}$ points, is shown versus $\alpha$, where the $S_{vN}$ distribution in BZ is shown in the inset (with also $\alpha = 0.25$).

We also note that in a recent quantum Monte Carlo simulation applied at charge neutrality[53] (i.e. even integer filling), a quantum valley Hall state is found at intermediate coupling for a specific choice of kinetic energy terms. Such a state is different from the QAH found at odd integer filling here as it preserves the TRS with helical valley edge modes and undergoes a first-order phase transition into intervalley coherent insulator at strong coupling, consistent with the exact results obtained in ref. [26].

**Quantized Hall conductance.** To reveal the topological properties in the large-$\alpha$ phase, we perform a flux insertion experiment on the cylindrical geometry (c.f. the inset of Fig. 3a) and compute the Hall conductance. We thread a $\phi$-flux along the cylinder by modifying the boundary condition $c_{\mathbf{R}+W(\mathbf{L}_1-\mathbf{L}_2)+\delta_\lambda} \equiv c_{\mathbf{R}+\delta_\lambda}$ to $c_{\mathbf{R}+W(\mathbf{L}_1-\mathbf{L}_2)+\delta_\lambda} \equiv e^{-i\phi} c_{\mathbf{R}+\delta_\lambda}$ for XC geometry and $c_{\mathbf{R}+W(\mathbf{L}_1-\mathbf{L}_2/2)+\delta_\lambda} \equiv c_{\mathbf{R}+\delta_\lambda}$ to $c_{\mathbf{R}+W(\mathbf{L}_1-\mathbf{L}_2/2)+\delta_\lambda} \equiv e^{-i\phi} c_{\mathbf{R}+\delta_\lambda}$ for YC geometry. During the process of the flux insertion, $\phi$ is adiabatically increased from 0 to $2\pi$ in the DMRG calculations. One thereafter obtains the Hall conductance $\sigma_H = \frac{e^2}{h} \Delta Q$ by measuring the net charge pumping $\Delta Q$ from one edge of the cylinder to the other. In DMRG, we calculate the net charge transfer as $\Delta Q = \sum_{x=L-l+1}^{L} [\tilde{n}_x^{col}(\phi) - \tilde{n}_x^{col}(0)]$, i.e. the pumped charge to the rightmost $l$ columns (chosen as $l = 3-4$ in practice) where $\tilde{n}_x^{col}(\phi)$ is the deviation of the charge number of the $x$-th column measured in the $\phi$-flux inserted ground state $|\psi_\phi\rangle$ from the half filling. For instance, we have $\tilde{n}_x^{col}(\phi) =$

$\sum_{y=1}^{W} \sum_{\lambda=1}^{2} \langle \psi_\phi | \hat{n}_{(x-1)\mathbf{L}_1+y(\mathbf{L}_1-\mathbf{L}_2)+\delta_\lambda} - \frac{1}{2} | \psi_\phi \rangle$ for the XC geometry, and similar expressions for YC.

As shown in Fig. 3a, for both XC and YC systems (with widths $W = 4$ and 6) in the QAH phase (e.g., $\alpha = 0.15$), we find a net charge transfer $|\Delta Q| = 1$ through a $2\pi$ flux insertion, showing that the Chern number $C = \pm 1$. In addition, Fig. 3b shows the column charge distribution $\tilde{n}_x^{col}$, where a half-charge $\pm \frac{1}{2}$ appears in two edges in $|\psi_{\phi=0}\rangle$. As $\phi$ gradually increases, the left/right-end charge smoothly reduces/increases from $\pm \frac{1}{2}$ to $\mp \frac{1}{2}$, which corresponds to an end-to-end pumping of a unit charge $\Delta Q = 1$, without "disturbing" the charge distribution in the bulk. We note that there is two-fold degenerate QAH ground state (apart from the additional degeneracy due to half-charge zero edge modes, see discussion below), and the charge pumping could be $\Delta Q = \pm 1$, corresponding to the spontaneous TRS breaking states with $C = \pm 1$.

**Understanding the TMI phase.** With DMRG calculations, we can also calculate the single-particle Green's function $G_{\lambda,\lambda'}(\mathbf{R} - \mathbf{R'}) = \langle c_{\mathbf{R}+\delta_\lambda}^\dagger c_{\mathbf{R'}+\delta_{\lambda'}} \rangle$, from which we can find the electron occupation $n_{\lambda,\lambda'}(\mathbf{k})$ in the momentum space. Due to the two-sublattice structure, $G_{\lambda,\lambda'}(\mathbf{R} - \mathbf{R'})$ and its Fourier transformation $\tilde{G}_{\lambda,\lambda'}(\mathbf{k})$ are both $2 \times 2$ matrices (cf., Supplementary Note 1). The two eigenvalues $\{n_1(\mathbf{k}), n_2(\mathbf{k})\}$ of $\tilde{G}(\mathbf{k})$ are shown in Fig. 3d. We find for all allowed $\mathbf{k}$ points, the larger eigenvalue $n_2(\mathbf{k}) \simeq 1$ and the smaller value $n_1(\mathbf{k}) \simeq 0$, representing the "two-orbit" electronic structure with one orbit filled while the other left empty. Albeit small, charge fluctuations between the two orbits are still present. We compute the von Neumann entropy $S_{vN}(\mathbf{k}) \equiv -\sum_{\lambda=1}^{2} n_\lambda(\mathbf{k}) \ln n_\lambda(\mathbf{k})$ that measures the deviation of the DMRG ground state from a Slater determinant of Bloch states. In Fig. 3e, we show the calculated $\bar{S}_{vN}$ averaged over the first BZ, which decreases as $\alpha$ increases, and becomes very small for large $\alpha$ cases. For example, we show the detailed $\mathbf{k}$-dependent profile for the $\alpha = 0.25$ case, in the inset of Fig. 3e. The relatively small $S_{vN}$ values suggest the QAH state, emerging in the interacting TBG model as revealed by DMRG calculations, actually very much resembles the Slater determinant ground state of the Haldane model and thus can be captured by a mean-field description.

To be specific, for small $\alpha$, a second-order perturbation shows the charging term $\sum_{\hexagon} (Q_{\hexagon} - 1)^2$ favors the insulating phases in which each hexagon of the honeycomb lattice contains exactly one electron, i.e. $Q_{\hexagon} = 1$ for every hexagon. Among all the states satisfying this requirement, the first- and second-order corrections from the cross terms $T_{\hexagon}(Q_{\hexagon} - 1)$ vanish. The stripe phase is selected from such states because it minimizes the contribution of $\langle \sum_{\hexagon} T_{\hexagon}^2 \rangle$, with the energy $\langle H \rangle_{stripe} \approx \alpha^2 U_0$ (c.f. Supplementary Note 2).

For large $\alpha$, motivated by the resemblance of the DMRG ground state to the Slater determinant, we perform a variational mean-field calculation that approximates the true ground state with the ground state of a tight-binding model containing various hoppings (see Methods and Supplementary Note 3). In particular the Fig. 1g demonstrates the emergence of NNN currents which constitute a loop in each hexagon, spontaneously choosing either the left- or right-chiral direction (here the right chirality). We find that the cross terms, i.e. $\langle T_{\hexagon}(Q_{\hexagon} - 1) \rangle_{QAH}$ become negative and thus favor the QAH phase. Including both the charging terms and $\sum_{\hexagon} T_{\hexagon}^2$, the variational mean-field analysis results in $\langle H \rangle_{QAH} \approx U_0(0.037 - 0.27\alpha + 0.71\alpha^2)$. Therefore, as $\alpha$ continuously increases from 0, the mean-field theory also finds the first-order

phase transition from the stripe phase to the QAH, in agreement with the DMRG result mentioned earlier. The mean-field energy is shown in Fig. 2a as indicated by the blue and red dashed line for the stripe and QAH phases respectively. Both lines provide a good approximation to the DMRG energy curve, and the intersection of two mean-field energies also provides a very good estimate of the QPT value $\alpha_c^{MF} \simeq 0.125$. Interestingly, the energy difference between the mean-field approximation and the DMRG calculation decreases as $\alpha$ moves away from the QPT, reflecting the suppression of the quantum fluctuations for large $|\alpha - \alpha_c|$, also illustrated by the $S_{vN}$ in Fig. 3e.

Moreover, as shown in Fig. 3b, there exist half-charge zero modes on both edges of the cylinder with even $W$, which also coincide with the Haldane model wrapped on the cylinder (for more details, see the Supplementary Note 4). We also compute the entanglement spectrum (ES), defined as $\xi_i \equiv -\ln(\rho_i)$ with $\rho_i$ the eigenvalues of the reduced density matrix. As shown in Fig. 3c, when we cut at the center of the system, a two-fold degeneracy in the ES is evident, which accounts for the half-charge zero modes in the edge (c.f. Fig. 3b), through the bulk-edge correspondence.

## Discussion

As we mentioned, the QAH can be obtained from narrow band models of TBG with large Coulomb interactions, but these models are built in the basis of extended states[27,28,43–45] making the interaction potential rather unwieldy. The results indeed show that several phases: QAH, strongly correlated topological semi-metal, and insulating stripe phases, are energetically competitive for the ground states at odd integer fillings[28,35,44,46,67].

The common belief, however, is that the nontrivial symmetry-protected topology of the narrow bands prevents a faithful construction of models within exponentially localized basis even when the bands' total Chern number vanishes[23]. On the other hand, as first shown in the context of the $Z_2$ topological insulators[68], the obstruction is not as severe as in the case of a nonzero Chern band (or band composite). If the total Chern number vanishes, the exponentially localized Wannier states can be constructed[69], but some of the protecting symmetries do not have a simple on-site implementation[68,70,71]. Because the transformation from the Bloch to Wannier basis is unitary and no information is lost in the process, it is therefore expected that the lattice tight-binding description should also result in the same ground state as found in unobstructed, extended states, basis. However, any practical implementation of this program needs to truncate the expansion of the interaction to on-site and few nearest neigbour sites. What is not obvious, therefore, is whether all the terms need to be included in the expansion or whether it can be truncated to recover the ground state.

The results presented here show that the truncation at just the nearest neighbor, parameterized by $\alpha$ in Eq. (1), is sufficient to recover the insulating and the topologically nontrivial phases. In addition, the main features of the single-particle excitation dispersion of the strong coupling correlated ground states at the charge neutrality point[53] from the model in Eq. (1) match those computed exactly in the extended basis[61,67]. This demonstrates the practicality of Wannier description even for such symmetry-obstructed bands. Our real-space interaction-only model therefore establishes the microscopic mechanism of the evolution between the insulating stripe and QAH phases. Our effective model and its unbiased numerical solution therefore revealed the essence of the physics in this particular regime, and is also consistent with other theoretical calculations[27,28,44,46].

As for relevance of our model towards the real system, it is understood that other than the $Q_O$ and $T_O$ terms, we do not include all the other projected interactions nor the small kinetic terms, i.e., the detailed feature of the TBG material, which will surely modify the specific value of $\alpha_c$. Apart from that they should not qualitatively alter the two phases and thus also the main conclusion of the present work. In addition to the ground states given above, the dispersion of the charged excitations produced by Eq. (1) is also found to be qualitatively consistent with more detailed calculation by two of the authors in refs.[61,63]. Reference[63] has also explicitly shown that the dispersion at the charge neutrality point is dominated by the $\alpha$ term in the chiral limit. For systems away from the chiral limit, it is expected that the inclusion of other terms may only quantitatively change the dispersion.

## Methods

**Density matrix renormalization group**. We employ the DMRG method, realized in the matrix product state form and with U(1) charge symmetry implemented, to accurately find the ground state of the TBG model. Following standard 2D DMRG calculations, we map the cylindrical geometries through a snake-like path, i.e., a quasi-1D structure, where highly controllable and efficient simulations can be performed. In practice, we retain up to $D = 512(1024)$ for $W = 4(6)$ cylinders, with truncation errors $\epsilon < 5 \times 10^{-5}$, for an accurate large-scale calculations. The detailed convergence check of the TBG model calculations can be seen in the Supplementary Note 1.

**Mean-field analysis**. We also applied the mean-field theory to approximate the interactions by a tight-binding model with variational hopping constants. The hopping amplitudes are obtained by minimizing the expectation value of the interactions in Eq. (1) for the state produced by the tight-binding model. In practice, the tight-binding model includes hopping amplitudes up to the 5th nearest neighbor. The details are presented in the Supplementary Note 3.

## Data availability

The data that support the findings of this study are available from the corresponding author upon reasonable request.

## Code availability

All numerical codes in this paper are available upon request to the authors.

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

## Acknowledgements

B.-B.C. and W.L. are indebted to Shou-Shu Gong, Xian-Lei Sheng, Xu-Tao Zeng, and Tao Shi for stimulating discussions. Y.D.L. and Z.Y.M. acknowledge the RGC of Hong Kong SAR of China (Grant Nos. 17303019, 17301420 and AoE/P-701/20), MOST

through the National Key Research and Development Program (Grant No. 2016YFA0300502) and the Strategic Priority Research Program of the Chinese Academy of Sciences (Grant No. XDB33000000). B.-B.C., W.L., and Z.C. acknowledge the support from the NSFC through Grant Nos. 11974036, 11834014, 12074024, and 11774018. O.V. was supported by NSF DMR-1916958, and by the National High Magnetic Field Laboratory through NSF Grant No. DMR-1157490 and the State of Florida. J.K. acknowledges the support from the NSFC Grant No. 12074276, and Priority Academic Program Development (PAPD) of Jiangsu Higher Education Institutions. We thank the Center for Quantum Simulation Sciences at Institute of Physics, Chinese Academy of Sciences, the Computational Initiative at the Faculty of Science and Information Technology Service at the University of Hong Kong, the HPC Cluster of ITP-CAS, and the Tianhe platforms at the National Supercomputer Centers in Tianjin and Guangzhou for their technical support and generous allocation of CPU time.

## Author contributions

Z.Y.M., W.L., and J.K. initiated the work. B.-B.C and Y.D.L. performed the many-body calculations. J.K. and O.V. conducted the theoretical analysis and mean-field calculations. All authors contributed to the analysis of the results. W.L., Z.C., and Z.Y.M. supervised the project.

## Competing interests

The authors declare no competing interests.
