## [Peer Review File · Nature Communications]

REVIEWER COMMENTS

Reviewer #1 (Remarks to the Author):

The author employ DMRG to study the lattice model describing twisted bilayer graphene introduced by some of the authors (Kang and Vafek). The main finding of the work is a phase diagram for the model in the single valley spineless limit at half-filling which is relevant to twisted bilayer graphene at filling -3 or $+3$. At the latter filling, experiments have already observed quantum anomalous Hall states as well as Chern zero states which is compatible with the two phases observed by the authors. The quantum anomalous Hall state has already been identified in several previous theoretical studies based on momentum space continuum model approaches including works by some of the authors (arxiv:2002.10360). The main novelty here is to be able to obtain this phase from a real space model. The paper is well written and represent an important addition to the growing literature on the topic. I recommend for publication provided the authors address a few comments that I detail below:

1. The authors make the claim that this is the first study to find a quantum anomalous Hall state. While this is likely true, a recent study by the same authors (Ref. 35) have already identified a strong coupling topological valley Hall state, which is a very close relative to the state identified here, using quantum Monte Carlo simulation of the same model at charge neutrality. It would be useful for the authors to clarify this point.
2. The paper will benefit from relating the model parameters to physical quantities, even on a qualitative level. For instance, most samples which observe the quantum anomalous Hall state are aligned with the hexagonal boron nitride (hBN) substrate. This induces a sublattice potential which breaks C_2 symmetry. The lattice model studied by the authors was derived for the pristine case without hBN alignment. What would change if this effect was included? is the α parameter going to increase or decrease? Similarly, it is unclear what physical quantities control the parameter α . Is it sensitive to lattice relaxation? to interaction screening?
3. The QAH spontaneously breaks two-fold rotation symmetry which is an emergent symmetry of twisted bilayer graphene. This symmetry is however implemented non-locally in the lattice model studied which means that any finite system will already introduce some explicit symmetry breaking that may slightly alter the energy competition by introducing a bias to C_2 -breaking states. While I do not expect any qualitative change to the phase diagram, I think the authors should comment on the issue and present some quantitative estimates for such explicit C_2 breaking due to finite size effects.

Reviewer #2 (Remarks to the Author):

In the present paper, the authors applied the state of art DMRG method to study the phases in twisted bi-layer graphene (TBG) based on a tight binding model constructed by the Wannier functions derived from a single valley. Overall speaking, I agree this is a timely research on a very hot topic in condensed matter by a reliable numerical method, which has been developed by some of the authors for years. Overall speaking, I agree that their paper has captured some features of the TBG system and deserves to be published somewhere. However, I don't think the present paper meets the scientific standard of nature communication due to an obvious weak point, the questionable relevance to the real materials. It is no doubt that TBG is an itinerant system and its Wannier orbitals for the flat bands actually extend to quite some range away from the center. In the theoretical model adopted by the present paper, only the interaction terms involving particles in the neighbouring super cells are considered in their DMRG calculation. Similar Wannier functions have also been discussed in detail in PHYSICAL REVIEW X 8, 031087 (2018). In table 1 of the above reference, the authors list the strength of the projected Coulomb interaction at different range, from which we can see that from on-site to the 5th neighbour the interaction strength only drops to about 1/3 of the on-site value. The above results suggest that in order to provide a faithful description for the correlation effects in TBG we definitely need to include further range interaction terms in the Wannier representation. Another problem of the strong coupling model discussed here is that the finite energy dispersion of the flat bands has been completely neglected, which I don't think is a good approximation. Comparing to the interaction strength U_0 (about 40meV) considered in the present paper, the energy dispersion of the flat bands (which is about 8meV at magic angle) is small but definitely not irrelevant. Based on the above two points, I don't agree to accept this paper for publication in Nature communication.

Reviewer #3 (Remarks to the Author):

In this work, the authors numerically and analytically study a local 'Hubbard-like' model on the hexagonal lattice which is derived from Wannier localizing the flat bands of twisted bilayer graphene. The Hamiltonian contains a single tuning parameter α , which is shown to drive a first order phase transition from a unidirectional charge density wave phase to a uniform quantum anomalous Hall state which spontaneously breaks time reversal symmetry. Within the field of moire materials the results presented in this work are of great interest because they show that even despite the fragile topology of the flat bands in the Bistritzer-MacDonald Hamiltonian, the electron interactions in the Wannier basis can be truncated to produce tractable real-space models which correctly capture the strong-coupling physics. The appearance of a quantum anomalous Hall phase in this model is also of general interest even after stripping away the connections to twisted bilayer graphene, because it is the first instance of a 'topological Mott insulator'. In his seminal paper, Haldane has shown that one does not need a magnetic field to have electrons occupy topologically non-trivial extended states which respond to Laughlin's adiabatic flux insertion by producing a quantized Hall current. The results in this work show that one does not even need a kinetic energy term in the Hamiltonian for this to occur.

The numerical results presented by the authors are detailed and convincing, and can in certain regimes also be matched to perturbative and mean-field calculations. This leaves little room for doubt about the correctness of the results. For this reason, I support publication of the manuscript.

There is perhaps only one statement in the paper that I would like to ask the authors to clarify. In particular, in the right column of page 3 it is stated that 'the NNN hopping amplitude is purely imaginary'. How is this hopping amplitude defined? And how do I see that it is indeed purely imaginary?

Nick Bultinck

Ref: NCOMMS-21-00509-T

Title: Realization of Topological Mott Insulator in a Twisted Bilayer Graphene Lattice Model

Dear Reviewers,

We thank you very much for the very positive assessment and insightful comments that have helped us to further improve our manuscript. We are more than happy to see that both Reviewers #1 and #3 have rather positive assessment of our work. They think our results constitute “an important addition to the growing literature” (Reviewer #1), and are “of general interest even after stripping away the connections to twisted bilayer graphene, because it is the first instance of a topological Mott insulator” (Reviewer #3). Besides, Reviewer #2 has raised some concern on the relevance to real materials. In the following, we give a point-by-point response to the comments, with the reviewers’ text being cited in blue and our subsequent response in normal format. Note that all bibliographic citations below make direct reference to the bibliography in our manuscript.

Overall, by taking the reviewers’ valuable comments and suggestions in to account, we have made careful revisions accordingly and believe our manuscript is now further improved.

Best regards,

Bin-Bin Chen, Yuan Da Liao, Ziyu Chen, Oskar Vafek, Jian Kang, Wei Li, and Zi Yang Meng

Response to Reviewer #1

Reviewer #1: The author employ DMRG to study the lattice model describing twisted bilayer graphene introduced by some of the authors (Kang and Vafek). The main finding of the work is a phase diagram for the model in the single valley spineless limit at half-filling which is relevant to twisted bilayer graphene at filling -3 or $+3$. At the latter filling, experiments have already observed quantum anomalous Hall states as well as Chern zero states which is compatible with the two phases observed by the authors. The quantum anomalous Hall state has already been identified in several previous theoretical studies based on momentum space continuum model approaches including works by some of the authors (arxiv:2002.10360). The main novelty here is to be able to obtain this phase from a real space model. The paper is well written and represent an important addition to the growing literature on the topic. I recommend for publication provided the authors address a few comments that I detail below:

Reply: We thank Reviewer #1 for his/her concise summary of our work and its relevance towards the recent developments of the field, we also thank Reviewer #1 for the support and the constructive comments.

Reviewer #1: 1. The authors make the claim that this is the first study to find a quantum anomalous Hall state. While this is likely true, a recent study by the same authors (Ref. 35) have already identified a strong coupling topological valley Hall state, which is a very close relative to the state identified here, using quantum Monte Carlo simulation of the same model at charge neutrality. It would be useful for the authors

to clarify this point.

Reply: We thank the reviewer for the valuable comment. Indeed, the quantum Monte Carlo study by some of us revealed the quantum valley Hall (QVH) state in the real space model at charge neutrality $\nu = 0$ for a specific choice of kinetic energy hopping terms. There are several differences between the QVH and the quantum anomalous Hall (QAH) state discovered in the present work. First, the two states are realized in different physical regimes. The QVH phase (Ref. [52] now in the updated reference list) is found in the intermediate coupling regime, and turns into the intervalley-coherence state (IVC) at the strong coupling limit of the model, so the QVH is more of theoretical interest than the realistic TBG experiment. In contrast, the QAH state discovered here is in the strong coupling limit at the filling of ± 3 , with both the spin and valley polarized. It should be closer to the real TBG system than that of the QVH. Second, these states are obtained in the strong coupling regime at very different fillings. As arXiv:2009.13530 illustrates, the QAH occurs at the odd fillings, and the ground state at the even fillings is usually the IVC state with zero Chern number.

We have incorporated these clarification into the revised manuscript as Reviewer #1 suggested.

Reviewer #1: 2. The paper will benefit from relating the model parameters to physical quantities, even on a qualitative level. For instance, most samples which observe the quantum anomalous Hall state are aligned with the hexagonal boron nitride (hBN) substrate. This induces a sublattice potential which breaks C_2 symmetry. The lattice model studied by the authors was derived for the pristine case without hBN alignment. What would change if this effect was included? is the alpha parameter going to increase or decrease? Similarly, it is unclear what physical quantities control the parameter alpha. Is it sensitive to lattice relaxation? to interaction screening?

Reply: We thank the reviewer for the insightful comment. Since the hBN alignment does not close the band gap between the flat bands and the remote bands, the Hilbert space of the flat bands are almost intact. Therefore, the Wannier basis and the associated projected Coulomb interaction are not affected by the alignment. However, the alignment breaks the C_2 symmetry and thus introduces the additional symmetry breaking terms in the Hamiltonian. Although such terms are small compared with the interaction and thus the system is still in the strong coupling regime, they favor the QAH state with a particular combination of the Chern number and the valley.

The parameter α controls the relative strength between the cluster charging and assisted hopping interactions. As mentioned above, the projected interaction including the value of α is not changed by the hBN alignment. As explained in Ref. [26], α is originated from the overlaps of two neighboring Wannier states, and therefore, it can be calculated by the Bistritzer-MacDonald model whose parameters vary with the lattice relaxation. More detailed discussions on the real space interaction model and the associated α will be presented in a separate work.

The form of the interaction in Eq. (1) is obtained when the screening distance is comparable with the moiré lattice constant, and thus contains only the cluster charging and the assisted nearest neighbor hopping terms. With the screening distance larger than moiré lattice constant, the interactions would include more terms for the sizable longer-range repulsion, but experimentally, barely change the qualitative properties of the insulating phases. Motivated by this observation, we argue our Hamiltonian in Eq. (1) provides a

minimal model for the explanation of the interaction dominated physics and the emergence of TMI.

We have incorporate these suggestions of the reviewer into the revised manuscript.

Reviewer #1: 3. The QAH spontaneously breaks two-fold rotation symmetry which is an emergent symmetry of twisted bilayer graphene. This symmetry is however implemented non-locally in the lattice model studied which means that any finite system will already introduce some explicit symmetry breaking that may slightly alter the energy competition by introducing a bias to C_2 -breaking states. While I do not expect any qualitative change to the phase diagram, I think the authors should comment on the issue and present some quantitative estimates for such explicit C_2 breaking due to finite size effects.

Reply: While the C_2 symmetry is broken by the truncation over the interaction range, the Hamiltonian in Eq. (1) is invariant under the combination of the chiral particle-hole, the unitary particle-hole, and $C_2\mathcal{T}$ symmetries. This combined symmetry can be locally implemented and belong to the $U(4) \times U(4)$ symmetry group in the chiral limit. Since this symmetry transforms the two QAH states with opposite Chern numbers into each other, it has to be broken for the system to develop a QAH state. Therefore, our model does not favor *a priori* the QAH state even if the C_2 symmetry is broken by truncating the interaction range. More detailed discussion on the real space model and the associated symmetries will be presented elsewhere in a separate work.

Response to Reviewer #2

Reviewer #2: In the present paper, the authors applied the state of art DMRG method to study the phases in twisted bi-layer graphene (TBG) based on a tight binding model constructed by the Wannier functions derived from a single valley. Over all speaking, I agree this is a timely research on a very hot topic in condensed matter by a reliable numerical method, which has been developed by some of the authors for years.

Reply: We thank the reviewer for the nice evaluation of our work, especially for his/her acknowledgement of our efforts in developing reliable lattice model of TBG and unbiased large-scale numerical methodologies in addressing them, over the years.

Reviewer #2: However, I don't think the present paper meets the scientific standard of nature communication due to an obvious weak point, the questionable relevance to the real materials. It is no doubt that TBG is an itinerant system and its Wannier orbitals for the flat bands actually extend to quite some range away from the center. In the theoretical model adopted by the present paper, only the interaction terms involving particles in the neighbouring super cells are considered in their DMRG calculation. Similar Wannier functions have also been discussed in detail in PHYSICAL REVIEW X 8, 031087 (2018). In table 1 of the above reference, the authors list the strength of the projected Coulomb interaction at different range, from which we can see that from on-site to the 5th neighbour the interaction strength only drops to about 1/3 of the on-site value. The above results suggest that in order to provide a faithful description for the correlation effects in TBG we definitely need to include further range interaction terms in the Wannier representation.

Reply: We thank the reviewer for the insightful comment and would need to go several different angles to address it.

First, we would like to point out that the reference PRX 8, 031087 (2018) (the reference [24] in our revised manuscript) has assumed that the Coulomb interaction is $\propto 1/r$ and projected it to the basis of the constructed Wannier states. Experimentally, however, the interaction is always suppressed at the long distance by the metallic gates. As a consequence, the long-range density-density interaction exponentially decays at the distance r exceeding the distance to the gates. Numerically, we have already found in the reference PRL 122, 246401 (2019) (Ref. [26] in our revised manuscript) that the projected interaction can be well approximated by Eq. (1) when the distance between double gates is about the order of the moiré lattice constant and this is case for the model in our present study.

Secondly, on the other hand, we completely agree with Reviewer #2 that in the TBG system the Wannier orbitals for the flat bands extend to quite some distance, and that is why over the past several years, we have been working very actively in developing reliable lattice model beyond the conventional on-site interaction and more importantly, developing unbiased numerical methodologies to gradually include more extended interactions into the computations. As the reviewer is well aware of, what have been presented in this work, with the third nearest neighbor interaction at the same strength of the on-site interaction, is already at the best of the present computational capability. While we are still developing better numerical methods that could take even longer range interaction into account in an unbiased manner, we would like to quote the words from Reviewer #3 to emphasize such an understanding, that, *“Within the field of moire materials the results presented in this work are of great interest because they show that even despite the fragile topology of the flat bands in the Bistritzer-MacDonald Hamiltonian, the electron interactions in the Wannier basis can be truncated to produce tractable real-space models which correctly capture the strong-coupling physics.”*

Thirdly, this work is primarily theoretical in that, although we tried to be qualitatively consistent with the TBG systems at $\nu = \pm 3$ fillings, our scope is actually beyond the TBG. We find our real-space model with interaction-only can give rise to the long-pursued topological Mott insulator state (i.e., the QAH state in the present case). This is certainly an important theoretical discovery in its own right, as such state has been investigated by the communities for more than a decade without success. We would again quote the text of Reviewer #3 to stress the point *“The appearance of a quantum anomalous Hall phase in this model is also of general interest even after stripping away the connections to twisted bilayer graphene, because it is the first instance of a ‘topological Mott insulator’. In his seminal paper, Haldane has shown that one does not need a magnetic field to have electrons occupy topologically non-trivial extended states which respond to Laughlin’s adiabatic flux insertion by producing a quantized Hall current. The results in this work show that one does not even need a kinetic energy term in the Hamiltonian for this to occur.”*

Reviewer #2: Another problem of the strong coupling model discussed here is that the finite energy dispersion of the flat bands has been completely neglected, which I don’t think is a good approximation. Comparing to the interaction strength U_0 (about 40meV) considered in the present paper, the energy dispersion of the flat bands (which is about 8meV at magic angle) is small but definitely not irrelevant.

Reply: We thank the reviewer for the comment. We believe the system is in the strong coupling regime for

the following two reasons. First, the original bandwidth $W \sim 8$ meV is much smaller than the interaction U_0 , and thus suggest the system is in the strong coupling regime. Second, in a recent work (Ref. [61]) by some of the authors, the wavefunction and the associated dispersion of the flat bands have been carefully calculated by integrating out the states on the remote bands. The corresponding superexchange interaction is found to be $\lesssim 10^{-2}e^2/(\epsilon L_m)$, that is tiny compared with U_0 and thus provides strong argument to neglect the kinetic terms. We have added such discussion in the revised manuscript.

Reviewer #2: Based on the above two points, I don't agree to accept this paper for publication in Nature communication.

Reply: With the detailed responses above and corresponding revisions in the manuscript, we humbly believe that our work both provides a microscopic mechanism of the observed QAH and, even more intriguingly, also goes beyond TBG by realizing the topological Mott insulator. We therefore sincerely hope Reviewer #2 can agree with us that it meets the standard of Nature Communications.

Response to Reviewer #3

Reviewer #3: In this work, the authors numerically and analytically study a local 'Hubbard-like' model on the hexagonal lattice which is derived from Wannier localizing the flat bands of twisted bilayer graphene. The Hamiltonian contains a single tuning parameter alpha, which is shown to drive a first order phase transition from a unidirectional charge density wave phase to a uniform quantum anomalous Hall state which spontaneously breaks time reversal symmetry. Within the field of moire materials the results presented in this work are of great interest because they show that even despite the fragile topology of the flat bands in the Bistritzer-MacDonald Hamiltonian, the electron interactions in the Wannier basis can be truncated to produce tractable real-space models which correctly capture the strong-coupling physics.

Reply: We thank the respected reviewer for his concise summary of our work and its relevance with the field.

Reviewer #3: The appearance of a quantum anomalous Hall phase in this model is also of general interest even after stripping away the connections to twisted bilayer graphene, because it is the first instance of a 'topological Mott insulator'. In his seminal paper, Haldane has shown that one does not need a magnetic field to have electrons occupy topologically non-trivial extended states which respond to Laughlin's adiabatic flux insertion by producing a quantized Hall current. The results in this work show that one does not even need a kinetic energy term in the Hamiltonian for this to occur.

Reply: Thanks for this insightful comment and deep understanding about the value of our work and the field of topological quantum matter.

Reviewer #3: The numerical results presented by the authors are detailed and convincing, and can in certain regimes also be matched to perturbative and mean-field calculations. This leaves little room for

doubt about the correctness of the results. For this reason, I support publication of the manuscript.

Reply: We thank Reviewer #3 for the firm support for the publication of our work.

Reviewer #3: There is perhaps only one statement in the paper that I would like to ask the authors to clarify. In particular, in the right column of page 3 it is stated that 'the NNN hopping amplitude is purely imaginary'. How is this hopping amplitude defined? And how do I see that it is indeed purely imaginary?

Reply: Thanks for the careful reading, and indeed we did not give the definition clearly enough. What we have calculated with the DMRG is not the NNN hopping constant but the equal time NNN correlation. On the other hand, the hoppings of the approximated tight binding model are obtained based on the variational analysis, as shown in the SM. We have added corresponding description in the revised manuscript, with the precise definition of NNN hopping now provided.

Summary of changes

All major changes are marked in blue color in the resubmitted text. Amongst them, main revisions are as follows.

- To respond the suggestions from Reviewers #1 and #2, on Page 2 we clarified the physical origin of the parameters in our model and discussed the irrelevance of the energy dispersion of the flat bands towards the strong coupling results revealed in this work, as well as the spontaneous nature of our QAH state without the need of including complex hopping terms to take the effect of hBN alignment into consideration.
- To respond the suggestion of Reviewer #3, on Page 3 (right column) we further explained the equal-time correlator $\langle c_l^\dagger c_{l'} \rangle$ that are found to be purely imaginary in the QAH state, and thus defined its imaginary part $\langle J \rangle = \frac{i}{2} \langle (c_l^\dagger c_{l'} - c_{l'}^\dagger c_l) \rangle$ as the order parameter of QAH phase.
- To respond the suggestion of Reviewer #1, on Page 4 (left column) we explained the difference between the QAH state discovered in this work and the QVH state discovered by some of us in a recent work.
- A number of references, including Refs. [9–20, 30–42, 47, 50, 53, 55–58, 63, 69], are updated and added to respect the fast development of the field.

REVIEWER COMMENTS

Reviewer #1 (Remarks to the Author):

The authors have satisfactorily addressed all my concerns. I have only a minor issue regarding a statement in the revised manuscript "In addition, we do not include the additional small symmetry-breaking term produced by the possible hBN alignment that favors the QAH phase with a particular Chern number". The estimate for this term is between 15-20 meV (1901.08110) which is around half the value of U_0 used in this work and can be larger if a different value of the screening epsilon is used, so it is not clear it can be dismissed as a small contribution. Maybe the authors should qualitatively discuss what is the likely effect of this term on the phase diagram they obtain.

Reviewer #2 (Remarks to the Author):

The authors clarified several points in the revised paper and their rebuttal letter. I agree with the authors that the real interaction is partially screened and can be expanded using the local forms as they did in this study. However, the authors also claimed that they have already included the local interaction terms up to the third neighbour, which I can't agree with. In the present paper, they take a very specific form of the local interaction, which contains "Q" and "T" terms. The problem is that these two terms can not cover all the local interaction terms when we expand the screened Coulomb interaction in real space using the Wannier orbitals up to the third neighbour. I don't think the authors have a strong argument that allow them to neglect all the remaining terms with the same order. For my second concern, I agree with the authors that the kinetic energy is much smaller than the interaction energy. However, the ground state of TBG is highly degenerate at the "flat" limit. There are many competing orders with the energy difference among them being around a few meV per electron. The kinetic energy is certainly at the same order of the energy difference between different competing order phases. Therefore, I don't think it can be neglected for a realistic model of TBG. Based on the above two points, I still can't agree to accept this paper for publication.

Reviewer #3 (Remarks to the Author):

The authors have satisfactorily addressed my comments. I can thus only repeat that based on the relevance and quality of the work, I support publication of this paper.

Ref: NCOMMS-21-00509-T

Title: Realization of Topological Mott Insulator in a Twisted Bilayer Graphene Lattice Model

Dear Reviewers,

We thank you very much for the very positive assessment and insightful comments that have helped us to further improve our manuscript. We are more than happy to see that both Reviewer #1 and #3 have rather positive assessment of our revised manuscript and we thank them for the recommendation of publication of our work. Reviewer #2 has raised some concern on the relevance to real materials. We have already pointed out that our effective lattice model cannot and does not need to include all the microscopic details of the real material in order to capture the main physics of the $\frac{3}{4}$ -filling TBG system, i.e. the topologically trivial correlated insulator and the quantum anomalous Hall state. Furthermore, our work acquires its own theoretical value by discovering the long-pursued topological Mott insulator for the first time (ever since the proposal by Raghu, Qi, Honerkamp and S.-C. Zhang in PRL 100, 156401 (2008), Ref. [47] in our manuscript), which as also pointed out by the Reviewer #3, such discovery has broader impact and fundamental theoretical value beyond TBG. We have further elaborated on these points in the reply below and in the revised manuscript.

In the following, we give a point-by-point response to the comments, with the reviewers' text being cited in blue and our subsequent response in normal format. Note that all bibliographic citations below make direct reference to the bibliography in our manuscript.

Overall, by taking the reviewers' valuable comments and suggestions into account, we have made careful revisions accordingly and believe our manuscript is now suitable for publication in Nature Communications.

Best regards,

Bin-Bin Chen, Yuan Da Liao, Ziyu Chen, Oskar Vafek, Jian Kang, Wei Li, and Zi Yang Meng

Response to Reviewer #1

Reviewer #1: The authors have satisfactorily addressed all my concerns. I have only a minor issue regarding a statement in the revised manuscript "In addition, we do not include the additional small symmetry-breaking term produced by the possible hBN alignment that favors the QAH phase with a particular Chern number". The estimate for this term is between 15-20 meV (1901.08110) which is around half the value of U_0 used in this work and can be larger if a different value of the screening epsilon is used, so it is not clear it can be dismissed as a small contribution. Maybe the authors should qualitatively discuss what is the likely effect of this term on the phase diagram they obtain.

Reply: We thank Reviewer #1 for his/her recognition of our efforts, and for the thoughtful comment. Indeed the symmetry-breaking term due to hBN alignment might not be small compared to U_0 . We have revised accordingly in the resubmitted manuscript.

Response to Reviewer #2

Reviewer #2: *The authors clarified several points in the revised paper and their rebuttal letter. I agree with the authors that the real interaction is partially screened and can be expanded using the local forms as they did in this study. However, the authors also claimed that they have already included the local interaction terms up to the third neighbour, which I can't agree with. In the present paper, they take a very specific form of the local interaction, which contains "Q" and "T" terms. The problem is that these two terms can not cover all the local interaction terms when we expand the screened Coulomb interaction in real space using the Wannier orbitals up to the third neighbour. I don't think the authors have a strong argument that allow them to neglect all the remaining terms with the same order. For my second concern, I agree with the authors that the kinetic energy is much smaller than the interaction energy. However, the ground state of TBG is highly degenerate at the "flat" limit. There are many competing orders with the energy difference among them being around a few meV per electron. The kinetic energy is certainly at the same order of the energy difference between different competing order phases. Therefore, I don't think it can be neglected for a realistic model of TBG. Based on the above two points, I still can't agree to accept this paper for publication.*

Reply: We thank the reviewer for pointing out the terms which we have not fully included in our TBG-related model. Indeed, we have not included all third neighboring interaction, nor the small kinetic terms, into consideration in the present work, based on the following reasons.

Firstly, this is because we have tried to extract the essential physics by using a model that is as simplified as possible to allow physical insight while keeping the integrity of the main non-trivial phenomenon. In this respect, for the $\frac{3}{4}$ -filling TBG system, our obtained phase diagram Fig. 1(f) of the effective TBG model contains two many-body phases: the translation-symmetry broken state and the QAH state. These two states are quite robust and more interestingly, are the only two types of the insulating phases obtained by various theoretical calculations (Ref. [27, 28, 44] and more recently arXiv:2105.05857 (Ref. [46])). The fact that the effective model and the unbiased numerical solution of the model have successfully captured these two phases, has already served the purpose of the effective model study of our work. Our calculation based on the Eqn. 1 has revealed the strong competition between the stripe and QAH states near the transition point $\alpha_c \simeq 0.12$. Other projected interaction and kinetic terms not considered here will surely modify the value of α_c , the detailed feature of the TBG material. Apart from that they should not qualitatively alter the two phases and thus also the main conclusion of our present work.

In addition to the ground states given in this manuscript, the dispersion of the charged excitations produced by Eqn. 1 is also found to be qualitatively consistent with more detailed calculation by two of the authors in Ref. [61] and [63]. Ref. [63] has also explicitly shown that the dispersion at the charge neutrality point is dominated by the α term in the chiral limit. For systems away from the chiral limit, it is expected that the inclusion of other terms may only quantitatively change the dispersion.

Lastly, we emphasize again that beyond its connection to QAH in TBG system, our work more importantly provide the first realization of topological Mott insulator in the strong coupling limit. The pursuit of this phase started ever since the proposal by Raghu, Qi, Honerkamp and S.-C. Zhang in PRL 100, 156401

(2008), Ref. [47] in our manuscript, and has not been succeeded until our work. It is therefore amazing and exciting, to Reviewer 3 and certainly also to us, that such a long-pursued interaction-driven topological state emerges from a concise and simple TBG lattice model considered in the present work. Such discovery has its broader impact and fundamental theoretical value to the communities of topological and strongly correlated physics. We sincerely hope that the respected Reviewer #2 would also appreciate such value of our work.

Response to Reviewer #3

Reviewer #3: The authors have satisfactorily addressed my comments. I can thus only repeat that based on the relevance and quality of the work, I support publication of this paper.

Reply: We thank the respected reviewer for firmly supporting the publication of our work.

Summary of changes

All major changes are marked in blue or light-gray color in the resubmitted text. Amongst them, main revisions are as follows.

- To respond the comment from Reviewer #1, on Page 2 (left column) we add the statement
“Although the projected interaction contains other terms such as next-nearest neighbor interaction, the more detailed calculations at the chiral limit have shown that the interaction induced dispersion of the charged excitation at the charge neutrality point is dominated by α , the nearest neighbor assisted hopping [63].”
- To respond the comment from Reviewer #1, on Page 2 (right column) we change the statement
“In addition, we do not include the additional small symmetry-breaking term [...] with a particular Chern number.”
 into
“In addition, we do not include the additional symmetry-breaking term [...] with a particular Chern number.”

- To respond the comment from Reviewer #2, on Page 2 (right column) we also add the sentence “*The kinetic term, as well as the further-range assisted hopping terms, may shift the critical value α_c of the phase transition but do not qualitatively change the phase diagram in Fig. 1(f).*”
- Typos are fixed and few new references are updated or added.